# Soil and Seed: Tumor Microenvironment Nurtures Immunotherapy Resistance and Renewal

**DOI:** 10.3390/ijms262110547

**Published:** 2025-10-30

**Authors:** Yiya Li, Qiushi Feng, Yangyang Xia, Lingzi Liao, Shang Xie

**Affiliations:** Department of Oral and Maxillofacial Surgery, Peking University School and Hospital of Stomatology & National Center for Stomatology & National Clinical Research Center for Oral Diseases & National Engineering Research Center of Oral Biomaterials and Digital Medical Devices, Beijing 100081, China; 2310124224@stu.pku.edu.cn (Y.L.);

**Keywords:** immunotherapy resistance, tumor microenvironment, immune checkpoint inhibitors, chimeric antigen receptor T-cell, overcoming strategies

## Abstract

Cancer immunotherapy has become a powerful clinical strategy for cancer management, while its efficacy is frequently limited by primary and acquired resistance. The tumor microenvironment (TME) plays a pivotal role in mediating such resistance through multifaceted mechanisms involving cellular, metabolic, mechanical, and microbial components. This review systematically examines how the TME contributes to immunotherapy failure. We compare resistance mechanisms common to both immune checkpoint inhibitors (ICIs) and chimeric antigen receptor T-cell (CAR-T) therapies, two cornerstone modalities in clinical practice. Furthermore, we discuss emerging strategies designed to overcome these barriers, including immune microenvironment, stromal normalization, metabolic modulation, and microbiota engineering. By integrating recent preclinical and clinical insights, this review aims to provide a comprehensive framework for understanding and targeting microenvironmental resistance, ultimately facilitating the translation of novel combination therapies into improved patient outcomes.

## 1. Introduction

Cancer immunotherapy has become a powerful clinical strategy for treating cancer, which aims to improve antitumor immune responses with fewer off-target effects. In cancer immunotherapy, agents activate the immune system to enhance its ability to target cancer cells using natural mechanisms, some of which are evaded as the disease advances. As shown in Figure 1, the number of approved immunotherapy drugs has been increasing in recent years, and a large number of therapeutic approaches are being developed in clinical and preclinical stages.

The early clinical immunotherapies for cancer include recombinant IFN-α and IL-2 [1,2,3]. While the initial clinical trials showed promise, advancements in cancer immunotherapy were hindered in the 200s primarily due to the unsuccessful outcomes of numerous vaccine trials. The first successful therapeutic cancer vaccine was approved for prostate cancer in 2010, with barriers to use. Shortly thereafter, a groundbreaking development was the approval in 2011 of ipilimumab, a monoclonal antibody targeting cytotoxic T lymphocyte antigen 4 (CTLA-4), for the treatment of advanced melanoma [4]. In recent years, innovative immunotherapies have emerged, such as checkpoint inhibitor monoclonal antibodies targeting programmed cell death 1 (PD-1) or its ligand, programmed death-ligand 1 (PD-L1) [5], alongside the pioneering chimeric antigen receptor T-cell (CAR-T) [6,7].

Immune checkpoint inhibitors (ICIs) function by alleviating the inhibitory signals that regulate T-cell (Tregs) activity [8], leading to heightened immune system activation and enhanced antitumor immune reactions. While T-cells have traditionally been central to ICIs therapy, the mechanism of action of ICIs extends beyond T-cell stimulation to encompass the activation of various innate and adaptive immune cells. These cells collaborate synergistically to mount a potent antitumor defense. The impact of ICIs on the immune response manifests not only within tumors but also in peripheral sites such as draining lymph nodes and peripheral blood [9].

CAR T-cell therapy, as a rapidly developing frontier in the field of cell therapy, has been gradually extended to hematological tumors, solid tumors, and some non-cancer diseases [10]. This technology combines adoptive cell infusion with advanced engineering strategies to achieve functional remodeling by extracting patient T-cells in vitro and introducing synthetic genetic elements. Nowadays, the CAR T-cell has undergone many generations, and its core development path focuses on enhancing the anti-tumor efficacy of T-cells by optimizing the structure of chimeric antigen receptors and constantly broadening the scope of targeted antigens. In the field of hematological tumors, in addition to the widely used CD19 and BCMA targets, CD20, CD22, and other targets have received increasing attention; CAR T for T-cell malignancies has also made progress in the direction of CD5, CD7, CD30, and other targets. In more challenging solid tumor treatments, CAR T-cell development based on tissue-specific antigens has made a series of breakthroughs, such as targeting IL 13R α2, EGFR/EGFRvIII for glioblastoma, and CLDN18.2 for gastrointestinal tumors. Meanwhile, the CAR T-cell design strategy, which has a bispecific antibody structure, has also become a hot research topic in the near future, providing a new direction for improving the efficacy of solid tumors [11,12].

Despite these advances, primary and acquired resistance to immunotherapy continue to limit overall clinical benefit. Many patients remain ineligible for ICIs or CAR T-cell therapies. Among those treated, responses are constrained by mechanisms such as suboptimal tumor specificity, low neoantigen expression, insufficient effector cell infiltration, and adaptive resistance processes [13]. In recent years, more studies have indicated that the tumor microenvironment (TME) in advanced solid tumors can hinder the penetration of therapeutic agents into tumors and suppress antitumor immune responses, thereby restricting the effectiveness of immunotherapy [14].

The TME is a highly structured ecosystem containing cancer cells surrounded by diverse non-malignant cell types, collectively embedded in an altered, vascularized extracellular matrix (ECM) [15].

In the TME, the proportion of cancer cells varies from 5 to 100% [16]. These cancer cells led to a low-pH environment with elevated lactic acid and decreased glucose levels, arising from the Warburg effect, which hostile microenvironment that directly impairs effector T-cell function while promoting immunosuppressive cell states. Key immunosuppressive cells include tumor-associated macrophages (TAMs), Tregs, and myeloid-derived suppressor cells (MDSCs), whereas cytotoxic T lymphocytes (CTLs) function as principal effectors. In addition, mesenchymal stem cells, endothelial cells, adipocytes, and neuroendocrine cells shape vascular tone, metabolic flux, and paracrine signaling, thereby influencing tumor behavior [17,18,19]. Communication between cells involves growth factors, chemokines, cytokines, inflammatory mediators, and other mediators, including exosomes, apoptotic bodies, circulating tumor cells, and cell-free DNA [20]. Through secreting the aforementioned factors, cells not only exert direct effects on tumor and immune cells but also modulate cellular activities, functional states, and aberrant paracrine secretion within the TME via intercellular crosstalk [21]. For example, MDSC-derived arginase depletes arginine to impair T-cell mitochondrial respiration [22]. Also, as a crucial component of the TME, ECM serves as a crucial context for cancer cells, offering both biochemical and biomechanical cues, which play a role in the mechanical microenvironment [21]. The ability of cancer cells to navigate through the ECM barrier, access the bloodstream, and form distant metastases is integral to cancer progression [23]. These processes collectively drive non-neoplastic cells to further remodel TME-centric niches, including the immune, metabolic, and mechanical microenvironments. Notably, with the development of microbial research, more and more studies have shown that microbial components in TME also play an important role [24]. Through dynamic remodeling of TAM-centric niches, the heterogeneous TME across distinct malignancies drives tumor progression while compromising immunotherapy efficacy, causing immunotherapy resistance [25].

In this review, we will systematically describe the multi-layered mechanisms by which TME regulates immunotherapy resistance, covering the functional state of various immune cells in the immunosuppressive microenvironment, intercellular communication network, accumulation of abnormal metabolites in the metabolic microenvironment and their molecular mechanisms, regulation of the mechanical microenvironment, and interaction between microbial community and host immune system. At the same time, we will comprehensively summarize the new intervention strategies for the above resistance mechanisms and integrate the latest research results to propose potential therapeutic targets, aiming to provide a theoretical basis and transformation direction for reversing immunotherapy resistance, improving overall response rate, and efficacy durability of immunotherapy.

## 2. Mechanisms Underlying Resistance to Tumor Microenvironment-Mediated Immunotherapy

TME is a complex system that can be divided into several subsystems. Most of the existing research focuses on immunotherapy resistance and its targeted overcoming strategies, mainly focusing on four subsystems: the immune microenvironment, metabolic microenvironment, mechanical microenvironment, and microbiota niche. Figure 2 systematically illustrates the intricate interactions between these four subsystems, which together contribute to multiple mechanisms that mediate immunotherapy resistance.

### 2.1. Tumor Immune Microenvironment

The composition of the tumor immune microenvironment (TIME) plays a key role in determining the strength of the initial immune response to the tumor and the effectiveness of the immunotherapy response [15]. The immunosuppressive microenvironment contributes to the development of immunotherapy resistance by reducing effector T-cells and increasing immunosuppressive cells, thereby weakening the killing effect of immunotherapy on tumors [14]. This microenvironment is shaped by a variety of immunosuppressive cells, either directly inhibiting effector T-cells’ function or secreting immunosuppressive cytokines. Moreover, these non-tumor cells do not function independently, but by building complex cell-to-cell communication networks, further exacerbating immunosuppression.

TAMs, as an important class of immunosuppressive cells, are highly plastic and their phenotypes change with environmental signals. TAMs polarized to the M1 phenotype when stimulated by IFN-γ or lipopolysaccharide and tended to differentiate to the M2 phenotype in the presence of IL-4, IL-13, and IL-10. M2-TAMs regulate inflammatory response and exert immunosuppressive function by secreting IL-10 and TGF-β [26]. In addition, they can secrete a variety of immune regulatory factors, such as cytokines, thereby weakening the effectiveness of immunotherapy. For example, a study based on monocytic ATAC and RNA sequencing found that highly activated TAMs overexpress SLAMF7 and PD-L1, simultaneously secrete CXCL9 and LGALS9, and recruit and inhibit T-cell function by binding to CXCR3 and TIM-3, respectively [27]. Also, activated M2-TAMs are capable of producing osteopontin, which mediates immunosuppressive responses through complex biological activities [28]. Similarly, MDSCs derived from tumor-related stress conditions directly remodel the immunosuppressive microenvironment by secreting a variety of cytokines, impairing effector T-cell and natural killer cells (NK cells) function, thereby driving immunotherapy resistance [22]. As a subset of CD4^+^ T-cells with strong immunosuppressive function, Tregs constitute an important immunosuppressive barrier in TME and promote therapeutic drug resistance. By depleting IL-2 and inducing TOX expression, Tregs directly lead to CD8^+^ T-cell depletion and cytotoxicity attenuation, and directly mediate immunosuppression by secreting TGF-β, which has become one of the key mechanisms of immunotherapy resistance [29]. The role of these immunosuppressive cells provides a theoretical basis for targeting these cells to remodel TIME.

In addition to immune cells, tumor angiogenesis and structural changes associated with microenvironment remodeling also change the secretion function of tumor-associated endothelial cells (TECs) during tumor progression, thus participating in the formation of immunosuppressive microenvironment. TECs inhibit T-cell function by secreting immunosuppressive factors such as IL-6, prostaglandin E2 (PGE2), indoleamine 2,3-dioxygenase, and IL-10, and by upregulating immune checkpoint molecules such as PD-L1 and B7-H3/4 [30]. Additionally, tumor-derived vascular endothelial growth factor A (VEGFA), IL-10, and PGE2 can induce TECs to express FASL, which in turn specifically clears effector T-cells through the apoptosis pathway. It is worth noting that high expression of cFLIP by Tregs can evade this killing mechanism, resulting in the rejection of effector T-cells and the accumulation of Tregs in TME [31].

Cytokines secreted by immune cells or stromal cells further regulate the shaping mechanism of the immune microenvironment. IL-4 is not only a key molecule for phenotypic regulation of TAMs [32] but also promotes immune escape and treatment resistance by downregulating TAP2 expression and remodeling chromatin status in its promoter region [33]. In addition, IL-10, another common cytokine, has been found to be involved in the formation of immunosuppressive microenvironment in tumors such as glioblastoma [34]. Meanwhile, the pleiotropic effects of IL-4, IL-10, and other cytokines have been revealed gradually with further study. For example, IL-10 has shown potential to promote anti-tumor immunity in a variety of mouse models, including by enhancing mitochondrial oxidative phosphorylation of macrophages, thereby enhancing the effectiveness of immunotherapy [35]. Based on these findings, researchers began exploring how to exploit the immune-enhancing properties of these cytokines. An engineered study of IL-10 showed that Fc-IL-10 fusion protein significantly prolonged IL-10 half-life in vivo and restored CD8^+^ T-cell function by regulating metabolic reprogramming of depleted T-cells. In solid tumor mouse models, Fc-IL-10 further demonstrated synergistic antitumor effects with adoptive T-cell therapy, with good safety and high efficacy [36]. Similarly, recent studies have found that IL-4, traditionally considered a type 1 cytokine, in its Fc fusion form (Fc-IL-4), as a type 2 cytokine, enhances CD8^+^ T-cell activity by activating the STAT6 signaling pathway, thereby enhancing the antitumor effect of adoptive cell therapy and ICIs, and this conclusion has been verified in various tumor models [37]. This provides a strong basis for restoring tumor sensitivity to immunotherapy by exogenous introduction of Fc-IL-4. In conclusion, Fc fusion cytokines exhibit important translational potential and clinical value in reversing tumor immunotherapy resistance. Future research can further explore the artificial engineering of IL-4, IL-10, and other cytokines to remodel their immune regulatory functions, thus expanding their application prospects in tumor immunotherapy.

In addition to IL-4, 10, other cytokines are involved in this process. Loss of the IFN signaling pathway is associated with tumor immune escape and poor prognosis after immunotherapy [38]. IFN-γ regulates PD-L1 expression on tumor cells and induces T-cell depletion. TGF-β, as an important factor secreted by Tregs and M2-TAMs, inhibits IFN-γ signaling by enhancing AKT-Smad3-SHP1 axis activity; blocking TGF-β in animal models reduces IFN-γ-induced Janus Kinase1/2 (JAK1/2) tyrosine phosphorylation and Signal Transducer and Activator of Transcription (STAT) activation, and inhibits the expression of immune checkpoint molecules such as PD-L1, thereby improving immunotherapy sensitivity [39]. In addition, highly expressed cytokines such as IL-8 and IL-33 also mediate immunotherapy resistance by promoting the aggregation of suppressive immune cells [14,40]. This suggests the potential clinical value of blocking inhibitory cytokines to block signaling pathways to remodel TIME.

Cytokines not only regulate killing activity directly but also affect recruitment, activation, and phenotype switching of immune cells by regulating intercellular communication. For example, TAMs promote the accumulation of MDSCs by secreting colony-stimulating factor 1 (CSF1) [26]. It is noteworthy that mononuclear MDSCs (M-MDSCs) migrating into TME can further differentiate into TAMs driven by hypoxia-inducible factor 1-alpha (HIF-1α) upregulation [41], reflecting the complexity of cell-to-cell communication and providing a new perspective for reversing immunosuppression. In addition, MDSCs promote the transformation of naive CD4^+^ T-cells into inducible Tregs and the expansion of natural Tregs by secreting IL-10 and TGF-β, which in turn feed back to enhance cytokine production in MDSCs, forming a positive feedback loop [29,42]. Stromal cells, such as cancer-associated fibroblasts (CAFs) and TECs, are also actively involved in cellular communication. AKAP12^+^ CAFs can induce monocytes to differentiate into immunosuppressive PD-1^+^ M2-TAMs and promote M1 to M2 phenotypic conversion, thereby increasing the M2/M1 ratio in immunosuppressive environments [43]. These M2-TAMs, in turn, activate CAFs in reverse, forming feedforward loops that exacerbate treatment resistance [44]. MYCT1-deficient endothelial cells promote cytotoxic T lymphocyte transendothelial migration and M1 macrophage polarization, which involves MYCT1 interacting with tight junction protein ZO-1 to regulate Rho GTPase-mediated actin cytoskeletal dynamics, thereby affecting endothelial cell movement in an angiogenic environment [45]. In addition, immune cells in TME can reshape the metabolic and mechanical properties of the microenvironment by interacting with ECM components. For example, immune cell-mediated ECM fibrosis physically represses effector T-cell infiltration, thereby promoting immunotherapy resistance [46]. These complex networks of cellular interactions provide new ideas for targeting specific immunosuppressive cells or multi-target intervention strategies and deepen understanding of the connections between subsystems within TME.

Multiple immunosuppressive cells and cytokines in TIME mediate the development of immunotherapy resistance through complex interactions. Current research has gradually expanded from targeting single-cell targets to intercellular communication networks and systematic remodeling of microenvironments, providing a solid theoretical basis for joint intervention strategies. However, this field is still limited by the high heterogeneity and dynamic evolution characteristics of TME. In the future, it is necessary to further explore the specific action mechanism of various cells and factors in different cancer species and disease stages by means of single-cell technology.

#### 2.1.1. ICIs

Based on various genomic analyses, numerous studies have shown that ICIs resistance is not only closely related to the TIME shaped by intercellular communication and inhibitory signals but also the formation of immunosuppressive microenvironment shows obvious tumor type and drug specificity. In different tumor types and different treatment regimens, non-tumor cells with specific phenotypes may mediate resistance mechanisms to ICIs through their biological activities.

In pancreatic ductal adenocarcinoma (PDAC), single-cell sequencing revealed that CD36^+^ CAFs recruit CD33^+^ MDSCs by interacting with macrophage migration inhibitor and its receptor CD74, thereby inducing resistance to anti-PD-1 therapy [47]. A similar mechanism exists in non-small cell lung cancer (NSCLC), where CAFs overexpressing COL13A1 recruit TAMs and Tregs by secreting chemokines and simultaneously inhibit dendritic cells (DCs) and cytotoxic T-cell infiltration [48].

In hepatocellular carcinoma (HCC), neutrophils expressing PLAUR enhance TAM-driven immunosuppression and cause depletion of CD8^+^ T-cells, leading to anti-PD-1 resistance. Notably, PLAUR inhibitors significantly enhance the efficacy of anti-PD-1 therapy in multiple preclinical models, demonstrating their potential value as a strategy to overcome resistance [49]. In addition, there are other neutrophil subsets in HCC that mediate treatment resistance, such as CD10^+^ ALPL^+^ neutrophils, which induce irreversible T-cell depletion through SELL-SELPLG-mediated cellular contact-dependent mechanisms, thereby impairing PD-1 blockade efficacy [50]. These findings suggest that multiple immunosuppressive mechanisms may exist even within the same cancer, providing a theoretical basis for targeting different cellular molecules in combination.

In addition, a phase II clinical trial in patients with peritoneal metastasis of gastric cancer showed that single-cell sequencing revealed a unique pattern of immunosuppression after sintilimab in combination with chemotherapy, including nivolumab or tirelizumab. Aggregated THBS2^+^ myofibroblast-like CAFs (mCAFs) recruit somatic tissue-resident macrophages via complement C3 and its receptor C3aR1 (C3aR1) and convert them into Secreted Phosphoprotein 1 (SPP1^+^) TAMs. These TAMs further form stroma-myeloid niches that support tumor growth and ultimately lead to the development of anti-PD-1 resistance [51]. These results further suggest that non-tumor cells in different tumor types may contribute to drug resistance through conserved immunosuppressive pathways, suggesting that targeting specific cell subtypes has potential for clinical application across cancer species.

In-depth study of non-tumor cells with specific functional subsets in different tumor types not only deepens the understanding of the heterogeneous formation mechanism of TIME but also provides a new direction for the use of molecular markers of specific cell subsets to assist precise treatment of ICIs. These studies lay a solid foundation for the development of novel therapeutic strategies and potential targets for reversing ICIs resistance, with important clinical translational value.

#### 2.1.2. CAR-T

Compared with ICIs therapy, CAR T-cell has significant differences in immunotherapy resistance mechanisms in hematological tumors and solid tumors, and their long-term survival and function maintenance ability in TME directly affects the invasion efficiency of CAR T-cells into tumor tissues, thus determining the therapeutic effect [52]. Current research focuses on the mechanism of action of specific immunosuppressive cells in CAR T-cell resistance and on the functional heterogeneity of the same cytokines in different tumor types.

In recent years, the mechanism of CAR T-cell resistance in hematological tumors has been studied continuously. A new study of patients with aggressive B-cell lymphoma identified a class of immune cell populations associated with CAR T-cell therapy resistance. This population exists in human and mouse models and is characterized as CSF1R^+^ CD14^+^ CD68^+^ LAMM cells, which significantly inhibit CAR T-cell function. Its mechanism of action is mainly through the PGE2-EP2/EP4 signaling axis by which T-cells directly interact, thus impairing the activity of CAR T-cells. The study further demonstrated in mouse models that anti-CD19 CAR T-cell combined with CSF1R blockers produced synergistic anti-tumor effects and significantly prolonged survival, providing a new strategy for clinical treatment of relapsed/refractory invasive B-cell lymphoma identified [53]. Additionally, a mononuclear single-cell multi-omics analysis based on 217 large B-cell lymphoma samples identified a class of tumor-derived exosomes prototypical patients whose TIME was characterized by the coexistence of hyperactivated macrophages and depleted T-cells. In such patients, depleted CD8^+^ T-cells activate macrophages by secreting IFN-γ and via IFN-γR. Super-activated macrophage markers PD-L1 and SLAMF7 are simultaneously overexpressed, and CXCL9 and LGALS9 are secreted, which recruit and inhibit T-cell function by binding to CXCR3 and TIM-3, respectively. The event-free survival rate at 12 months after CAR T-cell treatment in such tumor-derived exosomes prototype patients was only 39%, suggesting that the combination of PD-1 inhibitors or IFN-γ blockers may further enhance the efficacy of CAR T-cell in hematological cancer treatment by interfering with the immunosuppressive function of hyperactivated macrophages [27]. These patient-based single-cell-omics analyses not only reveal the unique role of specific immunosuppressive cell populations in different tumors and patients but also provide new potential targets and treatment directions for developing individualized treatment strategies for TME heterogeneity and improving CAR T-cell efficacy.

In addition to specific immunosuppressive cell populations, a variety of inhibitory cytokines also mediate immune tolerance by directly killing or functionally inhibiting CAR T-cells. Among them, the regulation of IFN family cytokines is particularly complex. For example, IFN-γ exhibits two-sided effects: acute exposure can enhance anti-tumor immunity, while sustained signaling may induce immunosuppression [54]. It is worth noting that in in vitro and in vivo experiments with hematological tumors, blocking IFN-γ signaling can reduce immune checkpoint protein expression and macrophage activation, and does not affect CAR T-cell efficacy but promotes its proliferation, suggesting that blocking IFN-γ in hematological tumors may enhance CAR T-cell function and reduce treatment-related toxicity [55]. In contrast, in solid tumors, loss of IFN-γR signaling pathway reduces the killing efficacy of CAR T-cells by increasing the resistance of tumor cells to CAR-T, mediating the development of CAR T-cell resistance in solid tumors. Gene level studies have also shown that deletion of IFN-γ pathway-related genes, including IFN-γR1, JAK1, and JAK2, significantly impairs CAR T-cell ability to clear solid tumors [56]. Therefore, maintaining moderate IFN-γ signaling in the TIME helps CAR T-cell function in solid tumors. Furthermore, CAR T-cells under chronic stimulation can mediate epigenetic inhibition through type I IFN signaling, resulting in dysfunction, and knockout of the transcription factor EGR2 can effectively reverse this inhibitory program [57].

CAR T-cell resistance mechanisms in blood tumors are closely related to specific immunosuppressive cells. However, the regulatory function of the same molecule in hematological tumors and solid tumors may be quite different. Therefore, care must be taken to differentiate between tumor types when restoring CAR T-cell sensitivity through targeted cytokine strategies. This finding is crucial for developing precise clinical treatment strategies.

Immunosuppressive cells and stromal cells construct cellular networks by secreting cytokines, leading to functional impairment and depletion of effector T-cells, thus mediating immunotherapy resistance. The difference in resistance mechanisms between ICIs and CAR T-cell stems from the heterogeneity of tumors and TME and is also related to the subtle differences in their therapeutic principles. This diversity highlights the need for individualized strategy optimization for different therapies, tumor types, and disease stages, while also providing potential therapeutic targets for overcoming resistance. Notably, shared immunosuppressive mechanisms between them provide a rationale for combination therapies.

### 2.2. Metabolic Microenvironment

In the tumor development, TME gradually forms unique metabolic characteristics due to numerous cell life activities such as proliferation of tumor cells and aggregation and infiltration of immune cells. Cancer cells ingest glucose in large quantities through the Warburg effect and perform aerobic glycolysis, while producing large amounts of lactic acid. Meanwhile, it mediates the depletion of other nutrients such as amino acids. Additionally, various metabolites of this abnormal metabolic microenvironment, including lactic acid, amino acids, fatty acids, etc., serve as signal molecules to further regulate metabolic reprogramming and intercellular communication of non-tumor cells, among which CAFs activation is closely related to the remodeling of the mechanical microenvironment, constituting a complex regulatory network of non-tumor cells, metabolic microenvironment, and mechanical microenvironment, which ultimately leads to the resistance of immunotherapy [58,59].

As a classical metabolic marker of cancer, lactic acid is mainly secreted by tumor cells and CAFs, directly promotes tumor growth through acidifying microenvironment, and mediates immunotherapy resistance by regulating life activities and metabolic reprogramming of non-tumor cells. For example, lactic acid can regulate TAMs activation, polarization, and metabolic remodeling. Lactic acid induces polarization of M2-TAMs via G protein-coupled receptors [60], which in turn exacerbates hypoxia and aerobic glycolysis, inhibits CD4^+^ and CD8^+^ effector T-cells with high metabolic demands, and promotes Tregs dependent on lipid oxidation [61]. In addition, lactate secretion promotes TAMs recruitment to TME through upregulation of the STAT3/CCL2 signaling axis and drives TAMs transcriptional reprogramming through the ENSA-STAT3 signaling pathway [62]. Activated TAMs release lipids through fatty acid metabolism, which causes lipid accumulation through the oxidized lipid-CD36 axis, gradually leading to T-cell dysfunction and mediating immunotherapy resistance [63,64]. In HCC, TAMs cross-present antigens to CD103^+^ CTLs via cytoplasmic pathways mediated by endoplasmic reticulum-related degradation mechanisms, resulting in CTLs remaining in the precancerous region and activating NLRP3 inflammatory bodies in macrophages, promoting tumor progression and immunotherapy resistance [65]. Similarly, lactate induces Treg accumulation and immunotherapy resistance through stabilizing the APOC2 structure by lactating it at K70, promoting extracellular lipolysis and free fatty acid release [66]. More importantly, lactic acid can be converted to acetyl coenzyme A, enhancing histone acetylation and directly mediating histone lactation, triggering epigenetic reprogramming and consolidating an immunosuppressive microenvironment [67,68]. In addition to indirect inhibition of effector cell function by remodeling the immunosuppressive microenvironment, lactic acid can also directly impair effector cell activity. For example, lactate secretion inhibits NFAT expression in CD8^+^ T-cells and NK cells, reduces IFN-γ production [69], and impairs NK cells function by inducing lactoyllysine modification, resulting in NAD metabolic disorders, mitochondrial disruption, and reduced cytotoxicity [70]. These results suggest that elimination or attenuation of lactate accumulation may serve as an important target for attenuating immunosuppressive microenvironments to restore immunotherapy sensitivity.

Among them, epigenetic reprogramming of NAD metabolism is one of the important mechanisms regulating tumor progression and immunotherapy resistance [71]. Nicotinamide N-methyltransferase (NNMT), as a key enzyme in the nicotinamide metabolic pathway, shows high expression heterogeneity in different tumors, especially in most gastrointestinal and urogenital cancers [72]. A metabolomic analysis revealed that NNMT is highly expressed in CAFs in lung adenocarcinoma. Because NNMT plays a central role in amino acid, organic acid, lipid, and nicotinamide metabolisms, its downregulation leads to an increase in intracellular methyl group levels, which promotes the upregulation of ECM remodeling-related genes in CAFs, enhances the invasion and metastasis of cancer cells, and is closely related to the shaping of the mechanical microenvironment.

In addition to lactic acid, hypoxia and abnormal cholesterol and amino acid metabolism induce effector T-cell depletion through a variety of mechanisms [73,74,75]. The hypoxia microenvironment is a basic characteristic of most malignant tumors. HIF-1α participates in cell proliferation and apoptosis, angiogenesis, glucose metabolism, immune response, and treatment resistance by regulating the expression of its target genes [75]. Specifically, HIF-1α can mediate PD-L1 expression to induce immune escape, thereby impairing the efficacy of immunotherapies, including ICIs and adoptive cell transfer [76]. A recent spatial transcriptomics study has also shown that HIF-1α regulates the physiological activities of immune cells. The HIF-1α complex binds to the ALCAM promoter, thereby increasing its expression in macrophages. These ALCAM-overexpressing macrophages co-localize with depleted CD8^+^ T-cells in the tumor spatial microenvironment and promote T-cell depletion, ultimately leading to immunotherapy resistance [77].

Similarly, abnormal lipid metabolism not only affects the behavior of cancer cells but also causes adaptive changes in lipid signaling in the surrounding TME. Lipid-driven phenotypic shifts in M2-type macrophages, CAFs, and CD8^+^ T-cells are closely associated with immunotherapy failure [63]. A clinical study of HCC showed that TAMs promoted ceramide-1-phosphate production (S1P) through high expression of NEK2, which promoted liver cancer progression and induced immunotherapy resistance. While blocking S1P weakened the immunosuppressive effect of macrophages, it increased the T-cell infiltration level and enhanced the immunotherapy effect [78]. Additionally, long-chain unsaturated fatty acids activate PPARγ through fatty acid binding protein 5 (FABP5), prompting TAMs to acquire immunosuppressive properties [79]. Another study targeting FABP5 showed that inhibition of the protein promoted iron death in tumor cells, upregulated CD80 and CD86 expression, and activated CD8^+^ T-cells [80], suggesting the feasibility of restoring immunotherapy sensitivity by regulating lipid metabolism. Similarly, FABP7 can be upregulated by HIF-1α to enhance DGAT1 activity in macrophages, which in turn delivers lipids to CD8^+^ T-cells and tumor cells via exosomes, resulting in CD8^+^ T-cell dysfunction and tumor cell proliferation [81]. The above findings suggest that targeting the FABP family to regulate abnormal lipid metabolism has important clinical translational potential and may become one of the important strategies to reverse immunotherapy resistance in the future.

Additionally, amino acid deficiency in the TME can lead to nutrient deficiency, and its concentration, transporter protein, and key enzyme activity also affect immune cell differentiation and function, thus mediating immunotherapy resistance [82]. Taking glutamine as an example, which is a key metabolic checkpoint between tumor cells and cDC1, while tumor cells compete for glutamine uptake through the SLC38A2 transporter, further impairing cDC1 function through the FLCN-TFEB signaling axis, thus inhibiting CD8^+^ T-cell killing activity [83]. Meanwhile, glutamine metabolites also mediate immunotherapy resistance in colorectal cancer (CRC). Through overexpression of its metabolite, glutamate, glutathione is associated with reduced T-cell invasion, and inhibition of this metabolic flux enhances tumor immunogenicity by activating ROS signaling [84]. Therefore, glutamine supplementation to restore CD8^+^ T-cell function, combined with inhibition of glutamate–glutathione flux, may become a key strategy to reverse immunotherapy resistance and provide new ideas for future clinical research.

In addition, tryptophan-kynurenine-AhR signaling pathway mediates immunotherapy resistance by regulating Tregs aggregation and differentiation. Tryptophan not only acts as an immunosuppressive factor on T-cells but also is decomposed into kynurenine, catalyzed by IDO [85], which is transported to T-cells via SLC7A5 or SLC7A8 and sensed by AHR, thereby upregulating PD-1 expression on CD8^+^ T-cells and promoting Tregs generation and differentiation [86,87]. CD8^+^ T-cell depletion and Tregs expansion ultimately lead to immunotherapy failure. The study also confirmed that AhR^+^ T-cells can differentiate into Tregs [88], further exacerbating the immunosuppressive microenvironment. These findings highlight the clinical value of targeting the tryptophan-kynurenine-AhR pathway to reverse drug resistance and restore immunotherapy sensitivity.

While regulating the life activities of non-tumor cells, the metabolic microenvironment itself is also reversely shaped by the cellular components in TME through the secretion of abnormal metabolites, which together contribute to immunotherapy resistance [89]. For example, MDSCs further exacerbate TME by producing reactive oxygen species, arginase enzymes, inducible nitric oxide synthase, and immunosuppressive cytokines such as TGF-β [22]. These factors can induce T-cell apoptosis, T-cell receptor nitrosation, and NK cell dysfunction, directly impairing the effect of immunotherapy [42].

It should be noted that different immunotherapies and tumor types can lead to heterogeneous metabolic remodeling of the TME. Non-tumor cellular components of the metabolic microenvironment also differ in their functional contribution mechanisms according to different immunotherapy strategies. Therefore, there is a need to further identify the unique metabolic characteristics in different tumors for different therapies in order to provide effective targets for overcoming treatment resistance.

#### 2.2.1. ICIs

The role of metabolic microenvironment in regulating ICIs resistance through abnormal metabolites and metabolic reprogramming of non-tumor cells has been further confirmed by experimental studies. The following highlights two distinct mechanisms of resistance, one relying on aberrant metabolites to block iron death and the other mediating metabolic reprogramming of non-tumor cells to trigger resistance.

One of the mechanisms of action of ICIs is to induce tumor iron death through CTLs. Itaconic acid in the metabolic microenvironment is associated with iron death failure. Itaconate, a metabolite produced primarily by activated macrophages catalyzed by ACOD1, can be taken up by tumor cells via the SLC13A3 transporter, thereby activating the NRF2-SLC7A11 pathway and protecting tumor cells from iron death. Further studies have shown that knocking out the ACOD1 gene in macrophages increases the sensitivity of tumor cells to iron death and improves the efficacy of nivolumab or pembrolizumab [90]. Another study revealed another mechanism by which itaconic acid mediates tumor immune escape: after itaconic acid enters tumor cells, it is alkylated with cysteine residue 272 of PD-L1 protein, thereby inhibiting its ubiquitination and degradation. This modification not only upregulates PD-L1 expression but also downregulates the expression of immunostimulatory molecules. Inhibition of SLC13A3 in syngeneic mouse tumor models synergizes with CTLA-4 blockade therapy to promote tumor-infiltrating T-cell immune responses, providing a potential novel therapeutic strategy for multiple cancer types [91]. These results suggest that targeting itaconic acid transporter SLC13A3 to block its uptake by tumor cells may restore therapeutic sensitivity to ICIs with significant clinical translational potential.

Furthermore, NNMT^+^ CAFs, mentioned above, have been shown to be significantly associated with non-response to PD-L1-blocking immunotherapy in patients with urothelial bladder cancer. Mechanically, NNMT^+^ CAFs recruit TAMs by mediating epigenetic reprogramming of serum amyloid A, driving tumor proliferation, and leading to resistance to neoadjuvant therapy with tirelizumab in combination with cisplatin and gemcitabine. Targeting NNMT significantly inhibited tumor growth and enhanced the apoptotic effects induced by anti-PD-L1 antibodies in mouse models of urothelial bladder cancer treated with the NNMT inhibitor 5-amino-1-methylquinoline iodide. These results further suggest that NNMT may be a potential target for overcoming resistance to anti-PD-L1 therapy [92,93].

The mechanism by which the metabolic microenvironment influences ICIs’ efficacy via non-tumor cell reprogramming and abnormal metabolite secretion remains unclear. However, investigating ICIs’ mechanisms, such as ferroptosis, and the roles of key metabolic enzymes across tumors will improve understanding of metabolic regulation in ICI response and identify new targets and strategies to overcome resistance.

#### 2.2.2. CAR-T

In addition to the restriction of TIME on the survival ability of CAR T-cells in solid tumors, the metabolic microenvironment also inhibits their substrate utilization through nutrient competition, thus impairing CAR T-cell proliferation, IFN-γ secretion, and cytotoxicity [68,94,95].

Traditionally, tumor cells have been thought to metabolize glucose to meet their proliferation needs, resulting in impaired effector T-cell function due to glucose deprivation in the TME [96]. It has been proven that tumor cells can significantly inhibit the function of tumor-infiltrating lymphocytes by competing for glucose in high-antigen TME [97]. Increasing glucose uptake by T-cells can enhance the immune response [98]. However, simply increasing glucose concentration in established tumors does not completely reverse the hyporesponsiveness of effector T-cells. Interestingly, the study also found that differences in glucose uptake between tumors were not necessarily associated with cell proliferation levels [97]. This seemingly contradictory phenomenon suggests that the mechanism of glucose restriction leading to T-cell dysfunction may not be universal, but relevant research has not yet thoroughly analyzed its underlying causes.

A recent breakthrough study systematically analyzed the uptake patterns of nutrients such as glucose, glutamine, and lipids by different cell types within tumors using multiple quantitative techniques. Although cancer cells consume about two-thirds of total tumor glucose, myeloid immune cells, especially TAMs and MDSCs, are most efficient at uptake at the single-cell level. In contrast, cancer cells dominate glutamine and lipid uptake. More interestingly, the preferential uptake of glutamine and fatty acids by cancer cells could reverse their own glucose utilization. Mechanically, this nutrient uptake preference is not determined by microenvironmental substrate concentrations but by intracellular programming regulated by the mTORC1 signaling pathway [99]. Due to the strong glucose uptake capacity of myeloid cells and the inhibitory effect of glutamine on glucose utilization by cancer cells, glucose is not actually scarce in the TME as traditionally thought, and energy restriction is not universal. Glucose uptake levels are actually determined by a combination of intracellular metabolic programs and glutamine uptake.

This new finding reasonably explains the aforementioned contradiction: differences in glucose uptake by different tumors not only depend on cancer cell proliferation but are also closely related to myeloid cell activity and tissue glutamine levels. This also explains why glucose supplementation alone cannot reverse T-cell depletion. Glucose deficiency is not widespread in the TME, and increased glucose is preferentially utilized by myeloid cells, rather than T-cells. Therefore, by engineering, such as overexpression of GLUT, specific enhancement of the T-cell uptake capacity can effectively improve the immunotherapy effect. However, increasing the glucose concentration alone had little effect because it could not change the competition pattern between cells.

These studies reveal the complexity of metabolic microenvironment regulation of T-cell function. It involves not only glucose competition between cancer cells and T-cells but also the interaction of multiple factors, such as metabolic characteristics of immune cells, intracellular regulatory programs, and nutrient levels, such as glutamine. Based on this, the dual strategy of moderately increasing glutamine to inhibit glucose uptake by cancer cells and combining engineered CAR-T, such as loading GLUT to enhance its metabolic adaptability, provides a new theoretical basis and promising intervention direction for overcoming CAR T-cell resistance. It should be noted that glutamine concentration needs to be precisely regulated, and excessive amounts can lead to organelle damage and T-cell depletion through ammonia toxicity, so dose balance is the key to effective intervention [100,101].

Moreover, hypoxia, another important feature of the metabolic microenvironment, regulates tumor-associated angiogenesis through the HIF-1α signaling pathway, thereby inhibiting CAR T-cell invasion into tumor tissues [102,103]. In glioblastoma, phosphoglycerate dehydrogenase-mediated endothelial cell metabolism promotes the formation of hypoxic and immunosuppressive vascular microenvironments, further demonstrating the close relationship between metabolic and mechanical microenvironments [104]. The specific mechanisms by which tumor blood vessels influence CAR T-cell infiltration are discussed in detail in the Mechanical Microenvironment section.

In addition to nutritional competition and invasion restriction leading to poor CAR T-cell efficacy in solid tumors, CAR T-cell resistance in hematological tumors is also associated with the metabolic microenvironment. A recent clinical study using single-cell RNA sequencing found a significantly higher proportion of macrophages overexpressing C1QB in the TME before CAR T-cell treatment in patients with refractory diffuse large B-cell lymphoma. Mechanistic studies have shown that M2 macrophages induce CD8^+^ T-cells to enter an immunosuppressive state through cholesterol efflux, thereby weakening CAR T-cell cytotoxicity and ultimately leading to its functional exhaustion [105]. The results highlight the value of sequencing technology in revealing CAR T-cell resistance and the metabolic mechanisms behind it, providing potential targets for developing novel sensitization strategies.

Metabolic microenvironment mediates the limited efficacy of CAR T-cells in solid tumors and hematological tumors through nutritional restriction, infiltration restriction, and immune cell metabolic reprogramming. It should be noted that, due to the more extensive clinical application of CAR T-cells in hematological tumors, the existing mechanism research focuses on specific pathways of specific refractory subtypes. While in solid tumors, relevant research still focuses on macro-mechanisms such as nutrient restriction, reflecting the importance of in-depth analysis of the hierarchical regulation mechanism of nutrient uptake for overcoming CAR T-cell resistance in solid tumors. Nowadays, remodeling the metabolic microenvironment by supplementing or reducing specific metabolites remains one of the important strategies for restoring CAR T-cell treatment sensitivity.

### 2.3. Mechanical Microenvironment

The tumor mechanical microenvironment has been recognized as an important emerging marker of cancer, characterized by increased ECM density, increased tissue stiffness, and increased solid stress [106]. Solid stress, fluid shear stress, and microstructural changes combine to impede immune cell infiltration [107,108,109]. Similarly, ECM also forms physical barriers by increasing stiffness, and its remodeling process can also promote the formation of immune escape microenvironments by transmitting physical signals such as mechanical forces and synergize with malignant features such as tumor microbiome, inflammatory state, and microvascular network, ultimately leading to immunotherapy resistance [106,110].

ECM remodeling is driven by multiple mechanisms, both directly by cancer cells and through cytokines or other non-tumor cells. As key stromal cells, CAFs drive ECM stiffening through cross-linking enzymes such as lipoxygenase and collagen [111]. Additionally, myeloid cells also contribute to ECM remodeling by producing remodeling-related enzymes, mediators, and ECM components [112].

These mechanisms lead to excessive deposition and crosslinking of collagen and hyaluronic acid in the ECM, significantly increasing ECM stiffness. Collagen density and tissue stiffness are key determinants of immune cell infiltration. Interstitial sclerosis and fibrosis caused by ECM remodeling physically limit the infiltration of cytotoxic T-cells and suppress anti-tumor immune responses, thereby contributing to immunotherapy resistance [106]. Studies have shown that T-cells tend to migrate along fine collagen and/or fibronectin fibers, avoiding dense matrices that are highly cross-linked [113]. Inhibition of lipoxygenase reduces collagen cross-linking, improves T-cell infiltration, and enhances anti-PD-1 therapy, suggesting that targeted ECM regulation has the potential to restore immunotherapy sensitivity [114].

In addition to directly affecting T-cell infiltration, ECM remodeling can also regulate the life activities of a variety of non-tumor cells, further promoting immunotherapy resistance. High stiffness ECM activates CAFs, promotes secretion of collagen, fibronectin, and glycosaminoglycans, and leads to matrix accumulation and fibrosis through downregulation of MMP-9 and upregulation of the tissue inhibitor of metalloproteinase-1 [110]. Stiffness changes can also enhance the glycolytic activity of CAFs, creating an immunosuppressive metabolic microenvironment [115]. In addition, DC maturation and macrophage function are also regulated by ECM mechanical signals. ECM proteoglycans affect the inflammatory phenotype, motility, and adhesion of TAMs through integrin signaling [116]. Under mechanical stress, TAMs activate TGF-β signaling pathways in response to mechanical stress, driving collagen biosynthesis. This process depletes arginine from the microenvironment while producing proline and ornithine, creating a metabolically inhibitory microenvironment that results in impaired CD8^+^ T-cell function and is closely associated with immunotherapy resistance, especially in female breast cancer [117].

Additionally, Piezo1, a mechanoreceptor present on CD4^+^ and CD8^+^ T-cells, can directly regulate T-cell function by sensing mechanical stress. Under conditions of increased matrix hardness, Piezo1 channel is activated and triggers calcium influx, which in turn initiates a calcium signaling cascade that promotes T-cell nuclear factor activation, ultimately leading to upregulation of PD-1 expression, accelerating CD8^+^ T-cell functional depletion, thereby mediating resistance to immune checkpoint inhibitors and CAR T-cell therapy [118]. Additionally, it also acts as a mechano-regulator in the immune microenvironment and promotes the expression of E3 ubiquitin ligase RNF114 by upregulating transcription factor GRHL3. RNF114 binds to filamentous actin, causing downregulation and recombination, which in turn inhibits T-cell traction [119]. This pathway may also mediate ICI resistance by affecting the antigen presentation function of DCs [120]. Meanwhile, studies have shown that targeting Piezo1 can restore the killing ability of effector T-cells [121], indicating the potential clinical value of targeting Piezo to restore immunotherapy sensitivity. In addition to direct regulation of T-cell function, Piezo1 may contribute to immunotherapy resistance through its involvement in collagen degradation and angiogenesis. For example, Piezo1 can enhance the activity of MMP-1 and MMP-10 and promote the degradation of collagen and fibronectin. This mechanism has been demonstrated to be closely related to the ability of tumor cells to detach from the primary lesion and migrate in high-grade serous ovarian cancer [122]. Additionally, in hepatocellular carcinoma, Piezo1 may be involved in tumor-associated angiogenesis driven by matrix stiffness by upregulating VEGF, CXCL16, and IGFBP2 [123].

Tumor blood vessels serve as key pathways for the delivery of ICIs, chemotherapy drugs, and CAR T-cells, and their functional integrity is critical to therapeutic response. Vascular structural abnormalities and dysfunction not only limit the infiltration of T-cells and drugs into the core of the tumor and induce local hypoxia but also further form a positive feedback loop between mechanical microenvironment and metabolic reprogramming, which exacerbates tumor malignant progression [124,125]. In addition to the mechanical sensing pathway relied on by Piezo1, increased ECM stiffness can also directly compress intratumoral blood vessels, affecting their morphological structure and physiological functions, and participate in the regulation of tumor angiogenesis [124]. Meanwhile, mechanical microenvironment can induce specific endothelial cell phenotypes, such as CXCL12^+^ TECs in HCC, which further enhance immunosuppressive microenvironments by recruiting MDSCs and inhibiting CD8 T-cell differentiation [126].

Non-tumor cells in TME regulate ECM remodeling by secreting collagen and proteases. ECM remodeling directly blocks immune cell infiltration by increasing collagen deposition and matrix stiffness and mediates non-tumor cell activity and T-cell dysfunction through mechanical signals while affecting tumor angiogenesis, thus forming a complex network of immunosuppression, mechanical, and metabolic microenvironments.

#### 2.3.1. ICIs

Since the overall response rate to ICIs is still unsatisfactory, the specific mechanisms of ECM remodeling and its regulation of immunosuppressive cell behavior have attracted increasing attention. Current research focuses on the abnormal expression of regulatory factors in ECM remodeling and the role of mechanical forces as physical signals.

Based on spatial transcriptome and metabolic analysis of the TME of patients before and after ICIs treatment, SPP1^+^ macrophages can interact with tumor cells by activating COL11A1^+^ CAFs, stimulating collagen fibers to deposit and entangle at tumor boundaries, thus hindering T-cell infiltration and ultimately leading to poor response of NSCLC patients to anti-PD-1 therapy [127]. However, this physical barrier of fibrotic ECM cannot completely prevent immune cell infiltration. Clinical data show that patients with fibroproliferative melanoma can still demonstrate high response rates to anti-PD-1 therapy [128].

Further studies have shown that ICI resistance also depends on ECM-mediated biological processes, including its regulatory effects on non-tumor cells [129]. For example, collagen deposition induces VEGFA production in tumor cells and promotes angiogenesis and endothelial cell proliferation by upregulating sex-determining region Y-box protein 18, thereby affecting tumor angiogenesis and mediating resistance to pembrolizumab [130]. In addition, collagen-fibrotic ECM may be involved in immunotherapy resistance mechanisms by regulating the function of immunosuppressive cells. A study in PDAC showed that collagen components can regulate PDAC energy metabolism, macrophage phagocytosis, and tumor growth via the Discoidin Domain Receptor Tyrosine Kinase 1 (DDR1)-mediated signaling axis, the collagen type I (Col1)-DDR1-NF-κB-NRF2 mitochondrial biosynthetic pathway, by activating the DDR1 receptor [131], providing a theoretical basis for targeted intervention in this pathway. Subsequently, another study demonstrated a similar mechanism: upregulated Col1 in the ECM induces the formation of immunosuppressive neutrophil extracellular traps via the Col1-DDR1-NFκB-CXCL8 signaling axis, further disrupting T-cell function and shaping an immunosuppressive microenvironment. The use of the DDR1 inhibitor nilotinib can reverse this immunosuppressive state and improve the efficacy of anti-PD-1 therapy [132]. These findings suggest that targeting DDR1, specifically by using DDR1 inhibitors to block the aberrant signaling axis triggered by upregulation of Col1, could attenuate the immunosuppressive microenvironment and thereby restore sensitivity to ICIs therapy.

Meanwhile, the increase in ECM mechanical force caused by collagen deposition itself can also play a regulatory role as a physical signal. Mechanical signaling is essential for monocyte differentiation and can be sensed by PYK2 through Piezo1 and integrin to promote F-actin polymerization and expression regulation of related genes, such as ACTR3 and RELA, in pancreatic cancer, thus promoting monocyte differentiation into macrophages. It has been further demonstrated that specific knockdown of PYK2 can disrupt differentiation and polarization of monocyte-derived macrophages and enhance the sensitivity of PDAC against PD-1 immunotherapy [133]. It is worth noting that mechanical forces can also enhance immunotherapy responses in certain contexts. For example, matrix rigidness-induced PD-L2 downregulation may promote iron death via the SMYD3 pathway, thereby enhancing the efficacy of anti-PD-1 therapy for HCC [134].

Both upregulated components of ECM remodeling and mechanical force signals can shape the immunosuppressive microenvironment by regulating immunosuppressive cell activity. These specific mechanisms of action provide potential strategies for remodeling normal ECM and alleviating ICIs resistance in different tumor types, and further clinical studies are worthy of assessing their safety.

#### 2.3.2. CAR-T

CAR T-cells have limited efficacy in solid tumors and hematological tumors due to their migration from the circulatory system to the lesion area, penetration of tissue barriers, and long-term maintenance of functional activity in the TME [95,135]. In this process, therapeutic resistance mediated by the mechanical microenvironment has an important correlation with pathological remodeling of the vasculature.

Driven by microenvironmental factors such as mechanical stress and hypoxia, key angiogenic factors such as VEGFA and FGF2 are induced and secreted in large quantities, participating in the formation of pathological neovascularization [136]. Although such factors maintain tumor nutrient and oxygen supply to a certain extent, persistent abnormal signals in the solid TME lead to uncontrolled angiogenesis and the formation of a structurally disordered and functionally defective vascular network, manifested in irregular lumens and pericytes, loss of cell coverage, and impaired barrier function [137].

These structural and functional abnormalities severely limit the infiltration efficiency of CAR T-cells. VEGFA interferes with the distribution of ICAM1 and VCAM1 on the endothelial cell surface by inducing nitric oxide production and impairs CAR T-cell adhesion and transendothelial migration ability [138]. FGF2 exacerbates endothelial dysfunction by inhibiting NF-κB nuclear translocation, making the local microenvironment more conducive to the aggregation of immunosuppressive cells, thereby affecting the functional activity of effector T-cells [139]. CAR T-cells are difficult to effectively infiltrate into the core area of solid tumors, which is one of the key mechanisms limiting their clinical efficacy.

Meanwhile, it has been found that VEGFA can also directly regulate the behavior of immune cells in blood tumors and participate in the formation of CAR T-cell therapy resistance. Studies in acute lymphoblastic leukemia patients and mouse models have shown that elevated VEGFA levels promote macrophage expansion and polarization toward the M2 phenotype, further exacerbating the immunosuppressive state. Notably, removal of macrophages or VEGFA knockout restored CAR T-cell efficacy. In GPR65 knockout mice, co-treatment with anti-VEGFA significantly prolonged survival [140]. Collectively, this evidence suggests that targeted VEGFA has important clinical value in restoring CAR T-cell therapy sensitivity in both solid and hematological tumors.

Additionally, dysfunctional vasculature fails to effectively transport oxygen and nutrients, exacerbating hypoxia and metabolic disorders within the tumor, further promoting immunosuppression and treatment tolerance. Its high permeability also provides a pathway for tumor cells to enter the circulatory system, increasing the risk of distant metastasis [141]. In solid tumors, disordered vascular structure hinders the effective infiltration of CAR T-cells and promotes the aggregation of inhibitory cells such as Treg through numerous mechanisms, such as forming physical barriers, interfering with T-cell migration and activation, and inducing apoptosis, to jointly maintain an immunosuppressive microenvironment.

It is worth noting that many immunotherapies represented by ICIs also rely on functional vasculature for drug delivery and T-cell infiltration. Therefore, normalization of vascular structure and function has become one of the strategies to improve the response to solid tumor immunotherapy. Nowadays, most of the research focuses on targeting key regulatory nodes such as VEGFA and exploring its joint application with armored CAR-T, immune checkpoint blocking, and metabolic regulatory drugs, aiming at reconstructing the perfusion and immune cell infiltration capacity of tumor areas [141]. Local intervention methods, such as intratumoral injection, chemical factor engineering, and matrix regulation, have shown preliminary efficacy in solid tumor types such as gastrointestinal tumors and sarcomas, providing feasible paths for overcoming tissue invasion barriers [142].

Mechanical microenvironment directly restricts the infiltration and function of CAR T-cells in solid tumors through multiple mechanisms, such as ECM fibrosis, vascular structure disorder, and increased interstitial pressure, and forms a positive feedback cycle with metabolic microenvironment to jointly promote immunotherapy resistance. It is noteworthy that local drug delivery, engineering, and combined matrix regulation have improved the infiltration efficiency of CAR T-cells in some solid tumor models, suggesting the potential of this strategy for future solid tumor immunotherapy. At the same time, targeting regulatory factors such as VEGFA, which play a key role in both solid tumors and hematological tumors, is expected to further expand the clinical application scenarios of CAR T-cell therapy and enhance its efficacy.

### 2.4. Microbiota Niche

With further research, the mechanism of action of microbiota as an active regulator of tumor progression and immunotherapy resistance has become increasingly clear [143,144]. The microbiota niche is a highly dynamic and complex ecological network composed of microbiota and its metabolites, including bacteria, fungi, viruses, archaea, and some parasites. Bacteria dominate the microbial community and participate in the regulation of the TME and the formation of immunotherapy resistance through various mechanisms, while fungi and viruses also play an important role in tumor progression and immunotherapy resistance [145,146,147]. It is noteworthy that the microbiota functions in a highly context-dependent manner, and its effects are influenced by the heterogeneity of the TME, presenting dual regulatory properties: promoting tumor progression and exerting inhibitory effects [148]. This seemingly contradictory phenomenon is supported by clinical data: in MSCLC, the presence of *Fusobacterium nucleatum* (*F. nucleatum*) is associated with poor response to ICIs treatment, while, in oral cancer, colonization of *F. nucleatum* is closely related to improved patient survival [149].

Microbiota often remodel TIME by secreting metabolites, and then construct a complex regulatory network connecting “metabolic-immune-microbiota microenvironment.” These metabolites can actively recruit immunosuppressive cells or aggravate the immunosuppressive microenvironment by regulating the metabolic state of immune cells, thus promoting the formation of immunotherapy resistance. For example, microbially derived bile acids can modulate RORγ^+^ Treg cells in the colon and recruit IL-17^+^ Treg, thereby promoting immune resistance [150,151]. Similarly, short-chain fatty acid butyric acid can promote Tregs proliferation by inducing CXCL16 expression, thereby enhancing the TIME [152]. In patients with advanced CRC, levels of 4-hydroxyphenylacetic acid, an important microbial metabolite produced by *Escherichia coli* and *F. nucleatum*, increased significantly. The metabolite upregulated CXCL3 expression by activating the JAK2/STAT3 signaling pathway, thereby promoting the infiltration of PMN-MDSCs in TME and inhibiting anti-tumor immune function of CD8^+^ T-cells. Studies have shown that knocking down CXCL3 restores tumor sensitivity to anti-PD-1 therapy [153]. Additionally, *F. nucleatum*-derived outer membrane vesicles contribute to immunotherapy resistance in head and neck squamous cell carcinoma (HNSCC) by altering tryptophan metabolism in TAMs, activating the TDO2/AhR pathway, and promoting transcriptional upregulation of immunosuppressive cytokines and checkpoint molecules [154]. *F. nucleatum* can also mediate immunotherapy resistance through other mechanisms. A recent CRC study showed that *F. nucleatum* of gut origin can drive CXCR1^+^ PD-L1^+^ phagocytes to migrate to tumors, further recruit PMN-macrophages via the CXCL2/8-CXCR2 and CCL5/CCR5 axes, promote tumor progression, and impair immunotherapy [155]. Given that *F. nucleatum* is closely associated with immunotherapy resistance in both colorectal and oral cancer, targeting this bacterium is considered a potential strategy for restoring immune sensitivity. It has been studied that PD-L1 antibody is loaded with engineered tumor symbiotic bacterial membrane nanovesicles to realize precise immunochemotherapy in a mouse CRC model [156], and the feasibility of reversing immunotherapy resistance with key bacteria, such as *F. nucleatum,* as targets has been verified.

It is worth noting that the key resistance-inducing bacteria represented by *F. nucleatum* have been confirmed to have a bidirectional regulatory effect on enhancing immune efficacy in many studies in recent years. In CRC models, a newer study has shown that by increasing the burden of *F. nucleatum* in tumors, it can increase the production of butyric acid in tumor cells, thereby inhibiting histone deacetylase 3/8 activity in CD8^+^ T-cells, promoting TBX21 expression, reducing PD-1 levels, and thus improving CD8^+^ T-cell function and enhancing anti-PD-1 therapy [157]. These contradictory results suggest that different tumor tissue invasion levels of *F. nucleatum* may mediate opposite immunomodulatory effects in the same tumor type by secreting different metabolites, and there are differences in immunotherapy resistance mechanisms in different tumor tissues, reflecting the profound influence of tumor heterogeneity on microbial regulatory function. Therefore, when targeting key microorganisms such as *F. nucleatum* to restore immunotherapy sensitivity, it is necessary to comprehensively consider its expression level in tissues and tumor heterogeneity background to formulate more precise clinical intervention strategies.

In addition to tandem metabolic and immunosuppressive microenvironments, bacteria are also involved in ECM remodeling that promotes immunotherapy resistance. Bacteria can cause ECM modification by regulating MMPs expression and actin cytoskeleton remodeling factors [158,159,160]. For example, elevated levels of proteobacteria trigger chronic inflammation, induce IL-1β release from PDAC cells, activate quiescent pancreatic stellate cells, and promote ECM deposition, thereby creating an inhibitory mechanical microenvironment that impairs CD8^+^ T-cell infiltration and function [161].

Additionally, antigen mimicry is also an important mechanism of bacteria-mediated immune resistance. Bacteroides fragilis express epitopes that cross-react with SOX4, leading to T-cell dysfunction [162]. Microbiota niche can also influence T-cell activation by interfering with antigen presentation, thereby promoting resistance to ICIs.

In addition to direct secretion of metabolites, bacteria can also regulate TIME by forming an interactive axis with fungi. Several studies have demonstrated that bacteria-derived short-chain fatty acids (SCFAs) are important regulators of fungal cell biology, inhibiting Candida albicans growth, reducing yeast-hypha morphological transition, and remodeling cell wall structure [163]. For example, in lung cancer patients, intestinal Candida overgrowth is often accompanied by an increase in lactic acid producers and a decrease in SCFAs producers [164]. Fungi themselves can also be directly involved in the construction of inhibitory TIME. A recent study of fungi in lung adenocarcinoma tumors found that *A. sydowii* promotes IL-1β-mediated myeloid suppressor cell expansion and activation via the β-glucose/Dectin-1/CARD9 axis in macrophages, leading to immunosuppressive TME and mediated immunotherapy resistance [165]. However, the exact mechanism by which bacteria activate fungi and remotely modulate resistance to immunotherapy remains unclear. To be sure, the bacteria-fungus-host dynamic interaction axis plays an important role in regulating tumor progression and immunotherapy response.

It is worth noting that, in addition to interactions between bacteria and fungi, viruses, as key components of microbial communities, also play an important role in immunotherapy resistance mechanisms. For example, EB virus, one of the important pathogenic factors of nasopharyngeal carcinoma, can target macrophages by secreting BRRF2 protein, interfere with the cGAS-STING signaling pathway, and then weaken their immune function, eventually leading to ICIs [166]. In addition to EBV, numerous other viruses have been implicated in immunotherapy resistance. For example, HPV, as an important cause of various tumors such as cervical cancer, penile cancer, and oral squamous cell carcinoma, can induce immunosuppression and thus weaken the effect of immunotherapy. It is generally believed that HPV, HBV, and other tumor-associated viruses promote immunotherapy resistance by mediating gene mutation and enhancing tumor heterogeneity. In recent years, research on relevant mechanisms has been gradually deepened. A newer study showed that the HPV E5 protein can suppress immune response by downregulating MHC expression and interfering with antigen presentation. Further experiments confirmed that HPV E5-expressing tumors were completely resistant to anti-PD-L1 treatment, and patients with high HPV16 E5 expression had poor survival [167]. Subsequent studies further revealed that HPV E5 can also promote the formation of stem cell-like phenotype in HPV^+^ HNSCC, which has low apoptosis, high activity, and strong tumorigenicity, thus mediating treatment resistance [168]. Heterogeneity of virus-infected tumors is also reflected in differences in tumor-infiltrating cells. A study of HBV-associated liver cancer showed that CCR4^+^ Treg were predominantly infiltrated in HBV^+^ tumor tissue, which expressed IL-10 and IL-35 more than CCR4^−^ Treg and was more potent at suppressing CD8^+^ T-cells, which led to resistance to sorafenib. Further experiments confirmed that blocking CCR4 can inhibit the infiltration of the Treg and restore the sensitivity of patients to sorafenib [169]. However, the specific mechanism of HBV-induced CCR4^+^ Treg aggregation remains to be clarified.

Meanwhile, in addition to the well-studied core species such as *F. nucleatum*, more and more studies have shown that other microbial communities also have two-way regulatory effects on immunotherapy, both mediating resistance and enhancing therapeutic effects. For example, the fungal metabolite verticillium A inhibits tumor growth and promotes immunotherapy response by selectively inhibiting histone methyltransferase, regulating cell cycle, apoptosis, stress response, and PD-L1 expression [170]. In liver cancer, Gram-positive bacteria upregulate CXCL16 expression in sinusoidal endothelial cells by mediating bile salt conversion, promoting NKT cell accumulation, and thereby enhancing immune response [171]. Analogously, *Finegoldii*-derived lipoprotein can enhance the expression of CXCL16 in CCR7^+^ DCs by activating the NF-κB signaling pathway and then promote CD8^+^ T-cell chemotaxis and invasion through the CXCL16-CXCR6 axis, effectively inhibiting tumor growth [172]. This mechanism not only provides a new idea for combined flora intervention and immunotherapy but also provides a reference for targeting CXCL16 to enhance T-cell recruitment.

As a key component of TME, the microbiota niche plays an important role in the establishment of immunotherapy resistance by regulating immunosuppressive microenvironment, remodeling ECM, and simulating antigens, as well as in the formation of a multi-domain interaction network involving bacteria, fungi, and viruses. Of particular significance is its dual functionality across various tumor types, treatment contexts, and levels of microbiota abundance. These observations not only comprehensively elucidate the intricate interplay between microorganisms, tumors, and the immune system but also offer novel collaborative targets and strategic avenues for surmounting immunotherapeutic resistance.

### 2.5. Others

In addition to the above mechanisms of inducing immunotherapy resistance, more and more studies have focused on the role of secretory vesicles in TME in the formation of immunotherapy resistance in recent years. At present, the research on secretory vesicles mainly focuses on two aspects: one is to explore the function of known components, such as PD-L1, and the other is to study the mechanism of novel immune regulatory factors, such as circular RNA.

As an important component of secretory vesicles, PD-L1 mediates immunotherapy resistance through multiple mechanisms. Exosomal PD-L1 binds directly to T-cell receptors and inhibits T-cell function [173]. In addition, a study in HNSCC showed that secretory vesicles carrying PD-L1 and adenosine-producing enzymes can influence T-cell infiltration by modulating neutrophil phenotype [174], leading to effector T-cell dysfunction and subsequent immunotherapy resistance. In addition to acting directly on immune cells to shape the immunosuppressive microenvironment, PD-L1 can also mediate immune tolerance by regulating metabolic pathways. For example, vesicular PD-L1 can cause DNA damage and abnormal activation of lipid metabolism in human and mouse T-cells, resulting in increased expression of lipid-metabolizing enzymes, accumulation of cholesterol and lipid droplets, and ultimately induce cell aging. Molecularly, vesicular PD-L1 promotes lipid metabolism by activating cAMP response element-binding protein and STAT signaling pathways [175]. These findings not only reveal the close regulatory relationship between extracellular vesicles and TIME and metabolic microenvironment but also suggest the potential value of targeting exosomes PD-L1 to remodel TME and improve immunotherapy sensitivity.

On the other hand, circular RNA (circRNA), as a new class of immune regulatory factors, regulates immunotherapy by various mechanisms, including assisting PD-1 function, weakening effector T-cell activity, and regulating tumor cell proliferation. For example, exosome-derived circCCAR1 can induce CD8^+^ T-cell dysfunction by stabilizing PD-1 protein, thereby promoting resistance to anti-PD-1 therapy [176]. Circ_0006896 has been shown to inhibit the efficacy of T-cell adoptive immunotherapy in acute myeloid leukemia both in vivo and in vitro. Its mechanism involves binding to histone deacetylase 1, reducing histone H3 acetylation, inhibiting transcription of genes associated with arachidonic acid metabolism, and thus disrupting CD8^+^ T-cell function and reducing the expression of cytotoxic molecules such as IFN-γ and granzyme B [177]. Similarly, circ_0001947, which is highly expressed in small extracellular vesicles (sEVs), upregulates CD39 by adsorbing miR-661 and miR-671- 5p, causing CD8^+^ T-cell depletion and immunotherapy resistance, and targeted blockade of this circRNA alleviates T-cell depletion and enhances response to PD-1 therapy [178]. In addition, circPETH-147aa secreted by TAMs drives methionine and leucine deficiency in CD8^+^ T-cells by enhancing HuR-mediated SLC43A2 mRNA stability, thereby impairing their function, and this circular RNA also enhances glycolysis and metastasis of liver cancer cells. Studies have shown that the compound Norathyriol can target this signaling axis, reverse the malignant phenotype described above, and synergize with anti-PD-1 therapy to enhance CD8^+^ T-cell toxicity, thereby overcoming ICI resistance [179].

Recent studies have further found that during CAR T-cell therapy, certain solid tumors respond to tumor necrosis factor secreted by CAR T-cells, which in turn promotes tumor cells to secrete small sEVs carrying tumor antigens. These sEVs, when taken up by surrounding CAR T-cells, can cause them to recognize their own antigens and trigger intercellular “fratricide”, which in turn causes CAR T-cell lysis and ultimately promotes the development of CAR T-cell therapy resistance [180]. This mechanism not only reveals the critical role of secretory vesicles as an important component of TME in modulating immunotherapy response but also highlights the significant clinical potential and significance of targeting tumor-associated vesicle secretion pathways.

Secretory vesicles in TME inhibit T-cell function, drive T-cell depletion, disrupt lipid metabolism, induce cell aging, and even trigger CAR T-cell self-destruction through antigen transfer by carrying PD-L1, circRNA, and other molecules, thus promoting immunotherapy resistance in many aspects. These mechanisms not only highlight the important role of vesicles in immunotherapy resistance but also provide a theoretical basis for their use as therapeutic targets.

## 3. Strategies to Overcome Clinical Resistance to Immunotherapy

The field of cancer immunotherapy has exploded, with strategies ranging from ICIs and CAR T-cell therapies to personalized cancer vaccines. However, immunotherapy resistance remains a major challenge limiting long-term survival. Current common strategies to overcome resistance include optimization of ICIs, combination of multiple inhibitors, and development of engineered CAR T-cells. For example, a Phase I clinical trial in Hodgkin lymphoma patients showed an optimal overall response rate of 53% (10/19) for the JAK inhibitor ruxolitinib in combination with the anti-PD-1 antibody nivolumab [181]. Other novel checkpoint inhibitors are also in clinical studies. In contrast, most of the engineering of CAR T-cells is still in the pre-clinical research stage and has not been widely used in clinical practice, and its feasibility needs more experimental verification [11]. The above contents have been summarized in Figure 3. Therefore, further exploration of potential new targets and targeted regulation of TME to improve treatment response will help improve patient survival and prognosis.

### 3.1. Targeting TIME

Remodeling immunosuppressive microenvironment to enhance immune response is one of the key strategies to restore tumor immunotherapy sensitivity. Common approaches include targeting immunosuppressive cells, blocking inhibitory cytokines and signaling pathways, and engineering CAR T-cells to enhance their long-term survival and functional maintenance in TME.

The immunosuppressive function of immunosuppressive cells can be weakened effectively by interfering with the recruitment and polarization process of immunosuppressive cells. CD11b agonists have been shown to induce phenotypic switching of TAMs, inhibit NF-κB signaling pathway, and activate the expression of IFN-related genes. This mechanism was demonstrated in tissue analysis in phase I trials, GB1275 enhanced antigen presentation by activating CD11b, which polarized macrophages towards a proinflammatory state, thereby promoting recruitment and retention of CD8^+^ T-cells [182]. Several ongoing clinical trials are evaluating the efficacy of TAM-targeted drugs in combination with ICIs. Among them, PLX3397 and its analogues in combination with PD-1/PD-L1 inhibitors have shown significant clinical potential. In glioblastoma, inhibition of ARPC1B reverses actin remodeling associated with M2-type polarization, reduces inhibitory cytokine production, and enhances response to PD-1 therapy [183]. Similarly, targeting Cd300ld, which is highly expressed on the surface of PMN-MDSC, has been suggested as a potential strategy for regulating recruitment and function of this cell [184].

In recent years, more and more research has focused not only on single targets but also on providing new perspectives on multi-target collaborative intervention. For example, simultaneous targeting of macrophages and neutrophils using the oncolytic virus OV-Mx40L effectively reshapes the immune microenvironment, a strategy that has been demonstrated to improve survival against anti-PD-1 therapy in mouse models [185]. These results fully demonstrate that targeting immunosuppressive cells has important clinical value and transformation prospects. In addition, important progress has been made in nanopharmaceutical research. A nanoparticle that can simultaneously target TAMs and MDSCs has been shown to reverse ICIs resistance in vitro and in vivo [186].

In addition to direct targeting of immunosuppressive cells, regulation of their related inhibitory cytokines and signaling pathways is also an important strategy to overcome ICIs resistance. Studies have shown that knockout of PPP2RA1 can regulate IFN-γ secretion, which in turn improves survival after immunotherapy [187]. Moreover, deletion or inactivation of the UBA1-STUB1 axis can stabilize JAK1, a key component of the IFN pathway, enhance IFN signaling, and upregulate the expression of important immune regulatory factors such as CXCL9, CXCL10, and MHC class I molecules, thereby enhancing anti-tumor immune response and effectively alleviating the limitation of ICIs resistance [188].

Similar to strategies used to overcome ICI resistance, CAR T-cell therapy can also be used in combination with drugs that target immunosuppressive components to improve survival after treatment. However, unlike ICIs, which rely primarily on reactivating endogenous T-cells, CAR T-cell therapy is critical to maintaining its function and persistence in the TME, which is full of inhibitory signals. CAR T-cells can be engineered to reduce phagocytosis by TAMs [189]. For example, by loading them with anti-phagocytic signaling molecules such as CD47 variants [189]. Similarly, the Smad signaling pathway can be disrupted by expressing dominant negative TGF-βRII, thereby maintaining CAR T-cell proliferation and cytotoxicity in a high TGF-β environment, having shown good efficacy and safety in mouse models [190]. In addition, CAR T-cells with IFN-γR knockout showed stronger tumor control effect, prolonged survival, and enhanced memory response in experiments, and were able to resist tumor re-attack, thus effectively alleviating CAR T-cell resistance constraints on treatment effect [191].

Targeting immunosuppressive cells, cytokines, and signaling pathways, and enhancing the survival and function of effector T-cells in TME to systematically remodel the immunosuppressive microenvironment has become an important direction to overcome immunotherapy resistance. This further highlights the importance of discovering new targets based on experimental studies and sequencing patient samples and driving their clinical translation.

### 3.2. Targeting Metabolic Microenvironment

Hypoxia and nutrient deprivation in metabolic microenvironments impair effector T-cell function, while abnormally accumulated metabolites such as adenosine and lactate inhibit TCR signaling and DC function. Therefore, current therapeutic strategies focus on reversing metabolic reprogramming of cellular components in the TME, increasing the production of beneficial metabolites while reducing the accumulation of inhibitory metabolites, thereby alleviating the inhibitory effects of the metabolic microenvironment on immunotherapy.

Lactic acid, as a typical abnormal accumulation metabolite in the metabolic microenvironment, has a signal transduction function besides metabolic waste. It promotes immunosuppressive programs by inducing histone lactylation modifications and upregulates the expression of immune checkpoint ligands [89]. Therefore, limiting lactic acid production and efflux is essential to regulate the metabolic microenvironment. Targeting lactate dehydrogenase, a tetrameric enzyme that catalyzes the interconversion of pyruvate and lactate, has proven to be an effective strategy. Oxaloate in combination with the PD-1 inhibitor pembrolizumab enhances CD8^+^ T-cell infiltration and inhibits tumor proliferation in a humanized mouse NSCLC model, thereby increasing sensitivity to immunotherapy [192]. Other strategies targeting lactic acid have also shown potential in combination with immunotherapy to overcome treatment tolerance [89].

In addition to regulating lactate metabolism, several studies have shown that the adenosine signaling axis centered on CD38 is easier to target clinically [193]. CD38 participates in adenosine production by consuming NAD, and targeting CD38 restores sensitivity to ICIs and elicits a stronger IFN response in human tumor explants and organoid models. In addition, the regulation of lipid metabolism has also received extensive attention. Lovastatin, as a cholesterol synthesis inhibitor, significantly enhanced tumor response to ICIs treatment in vitro and in vivo [194]. In a mouse model of oral squamous cell carcinoma, a simvastatin-loaded Ce6-PEG nanoplatform combined with anti-LDLR antibody (dose 1.0 mg/kg) further improved the immune microenvironment by modulating cholesterol levels and enhanced immunotherapy [195].

Regulation of lipid metabolism in myeloid cells also reduces immunosuppression [196]. Inhibition of fatty acid transport polarizes neutrophils to a cytotoxic N1 phenotype with a greater capacity for oxidative burst, showing synergistic effects with PD-1 blockade therapy in a melanoma postoperative model [196]. On the other hand, by knocking out LXRβ in CAR T-cells, cholesterol efflux can be inhibited, thereby enhancing its therapeutic effect on solid tumors [197].

CAR T-cells are prone to loss of mitochondrial membrane potential and depletion in hypoxic and hypoglycemic tumor regions, so engineering strategies focus on maintaining substrate supply and redox balance. For example, overexpression of GLUT1 to enhance glucose uptake and glycolytic flux helps maintain mitochondrial function and delays T-cell depletion, thereby enhancing its activity and clearance in tumors [198]. Similarly, overexpression of the FoxO gene can enhance mitochondrial adaptability and persistence of CAR T-cells and improve their therapeutic effect in vivo [199]. Several mouse model studies have demonstrated that CAR T-cells engineered to express IL-10 maintain mitochondrial activity and enhance oxidative phosphorylation [200].

By promoting essential amino acid uptake and reducing oxidative lipid stress, it can also delay T-cell function decline and maintain memory-like cell subsets in deep tumor regions. These metabolic intervention strategies complement ICIs therapy aimed at reducing levels of inhibitory metabolites in the TME [201].

Current research is focused on transforming the tumor metabolic microenvironment from an immunosuppressive state of hypoxia, acidification, and accumulation of inhibitory metabolites to a therapeutic response state of metabolic balance and restoration of immune function and enhancing T-cell function, as well as optimizing antigen presentation with engineered cells and nanomedicines. Such strategies, when combined with ICIs or CAR T-cell therapies, can drive microenvironment remodeling from “inhibitory” to “supportive,” providing a critical pathway for overcoming immunotherapy resistance.

### 3.3. Targeting Mechanical Microenvironment

As a physical barrier, the tumor’s mechanical microenvironment not only blocks the infiltration of effector T-cells by increasing interstitial fluid pressure, impairing blood perfusion, and exacerbating hypoxia but also actively participates in the shaping of the metabolic microenvironment. Normalizing the mechanistic microenvironment has thus emerged as a way to regulate the restoration of T-cell infiltration and function and to improve the metabolic microenvironment to improve immunotherapy survival. Current major strategies focus on targeting the ECM, including inhibition of profibrotic signaling pathways, direct degradation of fibrotic components, and regulation of key stromal cells such as CAFs to reshape the mechanical microenvironment.

Radiotherapy can act directly on ECM and CAFs, reducing their stiffness [202]. Several trials have demonstrated the effectiveness of radiotherapy in combination with immunotherapy, and available evidence suggests that immunotherapy may have the best effect when started early, concurrent with or after radiotherapy [203]. In addition, a novel nanoigniter and magnesium-peroxide-based bionic nanosystem (D/M-MP@LM) loaded with doxorubicin and metformin was designed to activate T-cell-mediated immunotherapy by comprehensively disrupting TME barriers. The nanosystem not only effectively initiates CD8^+^ T-cell immune response by promoting tumor antigen presentation but also significantly promotes T-cell infiltration by degrading rigid ECM. More importantly, with Mg^2+^ mediated metal immunotherapy, this system enhances the effector function of infiltrating CD8^+^ T-cells and avoids their depletion by improving the acidic microenvironment, thus fully activating T-cells and significantly suppressing tumors [204].

On the other hand, indirect intervention strategies for CAFs also show potential. For example, inhibiting autophagy of CAFs reduces IL-6 production, thereby disrupting the hyperfibrotic microenvironment and downregulating USP14 expression at the transcriptional level in pancreatic cancer cells [205]. Meanwhile, phase II clinical trials have shown that chemotherapy combined with immunotherapy can improve the sensitivity of anti-PD-L1 therapy by modulating the polarization state of CAFs and impairing their ability to produce fibrotic ECM [206]. In addition, novel CAR T-cell strategies such as armored CAR T-cells targeting vascular endothelial protein, CD248, can significantly inhibit tumor growth and metastasis by specifically attacking tumor-associated pericytes. E3K CAR T-cells showed specific killing activity against CD248-positive cells in three immunocompetent mouse models (BALB/c, FVB/N, and C57BL/6), effectively clearing target cells from the TME, resulting in increased tumor necrosis and growth inhibition [207].

By targeting ECM, combined with radiotherapy, nanodrugs, and new CAR T-cell technologies, it can effectively reduce matrix stiffness and promote T-cell migration and functional recovery, thus reversing immunotherapy resistance mediated by mechanical microenvironment. However, since some of the treatment regimens are still in the early clinical or even preclinical stage, their safety and effectiveness in humans need to be further verified. In the future, it is necessary to optimize and individualize the treatment strategy more accurately in combination with the specific biological characteristics of TME.

### 3.4. Modulatory Role of the Microbiota Niche

Given the dual role of microbiota niche in immunotherapy response and the ongoing research, more and more strategies are focusing on enhancing immunotherapy effectiveness through microflora transplantation and supplementation of specific species and their metabolites.

Fecal microbiota transplantation (FMT), as a common strategy, has been proved to be effective. Recent studies have shown that FMT in combination with PD-1 inhibitors provides clinical benefit and demonstrates good safety in patients with refractory solid tumors [208]. A large-scale follow-up study of patients with advanced melanoma further showed that FMT in combination with PD-1 blockers significantly prolonged survival [209]. Key to this strategy include successful colonization of donor strains, increased microbial diversity, and enrichment of immunostimulatory bacteria. At the same time, upregulation of antigen processing related genes and IFN signaling pathway transcripts can be observed in tumor tissues and peripheral blood [208,209], indicating that FMT can restore immunotherapy sensitivity by increasing immune response level.

In addition, metabolites of certain species have also been shown to enhance immunotherapy responses. SCFAs, as key metabolites of microorganisms, are closely related to the occurrence and development of CRC. In mouse and human models of CRC, elevated SCFAs enhance response to chemotherapy and immunotherapy [210]. Bifidobacterium-derived extracellular vesicles have been found to potentially modulate the efficacy of anti-PD-1 therapy for NSCLC [211]. These findings support the idea of adding specific species to improve treatment outcomes, but more clinical trials are needed to verify safety.

In CAR T-cell therapy, the modulation of the microbiota plays a crucial role in enhancing treatment efficacy by facilitating T-cell infiltration into tumor tissues and maintaining their functional integrity. One illustrative example involves the supplementation of *Akkermansia*, a beneficial gut bacterium, which has shown promise in enhancing the functionality of CAR T-cells in patients deficient in this microbe [212]. Additionally, the presence of butyrate-producing bacteria has been found to mitigate CD8^+^ T-cell depletion by inhibiting histone deacetylase activity. This mechanism is anticipated to bolster the anti-tumor effects of CAR T-cell therapy, particularly in “immuno-cold” tumors [157]. These microbiome-targeted intervention strategies help reshape the TME, thereby improving CAR T-cell recognition and affinity for target cells.

Microbial interventions show important potential to reverse tumor immunotherapy resistance. Immunotherapy sensitivity can be effectively restored through strategies such as FMT, specific probiotics, and their metabolites. In the future, precise intervention targeting the microbiome is expected to become an important direction to improve treatment resistance and improve patient survival.

## 4. Discussion

The role of TME in immunotherapy resistance has received increasing attention. This paper systematically expounds the mechanism of immunotherapy resistance mediated by four subsystems of TME, TIME, the metabolic microenvironment, mechanical microenvironment, and microbiota niche, and the relationship among these subsystems. Meanwhile, we explored the heterogeneity of TME caused by different tumor types and treatment approaches and the role of this heterogeneity in the development of unique resistance mechanisms to ICIs and CAR T-cell therapies. Based on our understanding of the mechanisms involved, we further propose potential strategies that might restore immunotherapy sensitivity. It should also be noted that there is significant heterogeneity in TME between patients, even within the same tumor type and therapy. In the future, it will still be necessary to use sequencing technology to classify and classify patients more finely, so as to mine patient prototypes with similar microenvironment characteristics.

In the process of sorting out the above immunotherapy resistance mechanism, we found that some key molecules play a tandem regulatory role in the four subsystems of TIME, metabolic microenvironment, mechanical microenvironment, and microbiota niche. Therefore, we propose innovatively that interventions targeting these pivotal molecules may overcome the complex compensatory mechanisms between TME subsystems that traditional therapies have difficulty coping with, thus breaking through the current limitations of unsustainable therapeutic effects. For example, targeting the AhR signaling pathway could simultaneously block CD8^+^ T-cell depletion mediated by the tryptophan kynurenine-AhR axis and inhibit the immunosuppressive effects of the AhR pathway in TAMs [86,87,88,154], while intervention with Piezo1 could synergistically restore normal ECM structure and reverse T-cell depletion and decreased migration caused by this mechanosensory pathway [121,122,123], thereby restoring immunotherapy sensitivity in a more systematic manner.

In view of the complex problems such as immune cell composition, cytokine function, and local nutrition distribution caused by tumor heterogeneity and individual differences of patients, a convolutional neural network (CNN) can be used to integrate multi-channel information [213], combine the spatial topology characteristics of TME with multi-group data, and thus simulate the functional state of TME under the joint action of multi-dimensional characteristics. Although the construction of such CNN models is highly dependent on the support of large-scale, high-quality datasets, their establishment is expected to build personalized “TME digital twins” for each patient to predict optimal treatment strategies and joint programs. Meanwhile, with the help of interpretable artificial intelligence technology, it is also expected to mine new treatment targets located at key nodes of the TME regulatory network from the model “black box” and provide an efficient virtual test environment for evaluating the safety and effectiveness of new targets and joint strategies.

Although the mechanism of TME-mediated immunotherapy resistance is complex, it contains clear intervention logic and transformation potential. By targeting pivotal molecules linking different subsystems and integrating multi-omics data using computational models such as CNN, we are expected to gradually overcome the therapeutic bottlenecks caused by tumor heterogeneity and tumor microenvironment compensation mechanisms. Future research should focus on building more accurate dynamic models of TME and driving the development of individualized therapeutic strategies to achieve substantial breakthroughs in reversing immune tolerance.

## Figures and Tables

**Figure 1 ijms-26-10547-f001:**
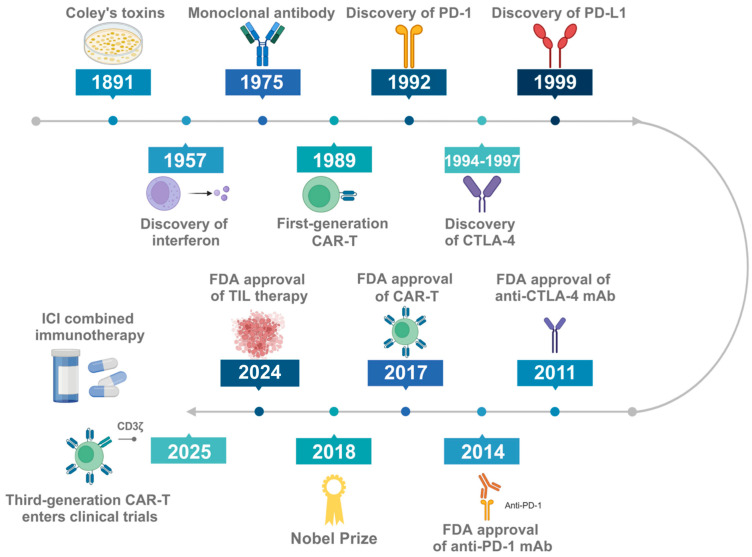
Milestones in the development and prospects of immunotherapy.

**Figure 2 ijms-26-10547-f002:**
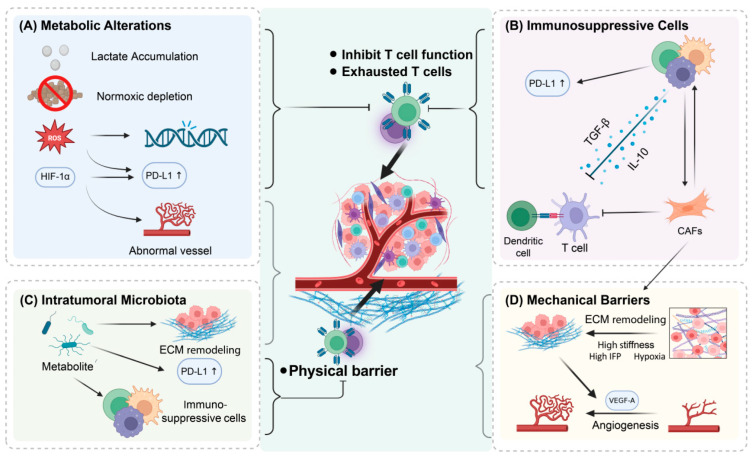
Key mechanisms mediating immunotherapy tolerance. (**A**) Metabolic alterations—such as lactate secretion, adenosine accumulation, and lipid droplet formation—impair T-cell function and promote epigenetic modifications (e.g., histone lactylation). (**B**) Immunosuppressive cells and CAFs interact via cytokine networks to inhibit cytotoxic T-cells and promote an immune-excluded phenotype. (**C**) Intratumoral microbiota and their metabolites modulate local immune responses through mediating the activity of immunosuppressive cells and PD-L1 expression, and antigen presentation. (**D**) Mechanical barriers, including ECM stiffness and abnormal vasculature, physically limit T-cell infiltration.

**Figure 3 ijms-26-10547-f003:**
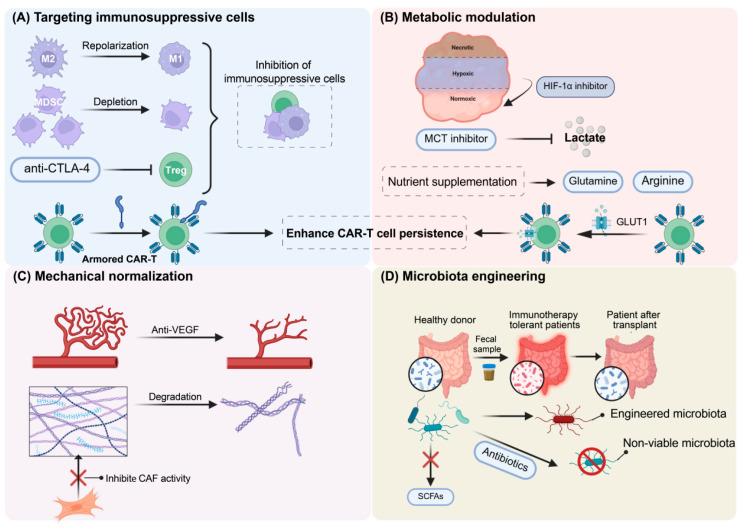
Emerging strategies to overcome TME-mediated immunotherapy resistance. (**A**) Targeting immunosuppressive cells: using CSF1R inhibitors or CD11b agonists to reprogram TAMs, depleting MDSCs or Tregs. (**B**) Metabolic modulation: inhibiting LDHA or CD38 to reduce lactate and adenosine, using statins or LXRβ knockout to modulate cholesterol. (**C**) Mechanical normalization: depleting CAFs, degrading ECM components, or normalizing tumor vasculature to enhance T-cell infiltration. (**D**) Microbiota engineering: employing FMT or specific probiotics (e.g., *Akkermansia*) to reshape the microbial niche and improve immunotherapy response. These combination strategies aim to reverse the immunosuppressive TME and enhance the efficacy of ICIs and CAR T-cell therapy.

## Data Availability

No new data were created or analyzed in this study. Data sharing is not applicable to this article.

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
