# Peer review of "Soil and Seed: Tumor Microenvironment Nurtures Immunotherapy Resistance and Renewal"

_ijms, 2025, doi:10.3390/ijms262110547_

Round 1
Reviewer 1 Report
Comments and Suggestions for Authors
This review summarizes resistance mechanisms in TME to immunotherapies, including immune checkpoint inhibitors and CAR-T. It nicely covered mechanisms in the aspects of metabolic alterations, immunosuppressive cells and physical barrier. Furthermore, the review also discussed the current strategies developed to overcome these barriers. The review should be published if the following questions can be addressed.
- In Figure 1, the FDA approved the first TIL (tumor-infiltrating lymphocyte) therapy — lifileucel (Amtagvi) — on February 16, 2024, instead of 2019.
- In 2.1. Tumor immune microenvironment, immunosuppressive cytokines, including IL4 and IL10 were discussed. However, recently IL4 and IL10 were reported to enhance CD8 functions in TME and therefore enhanced the efficacy of CAR-T and ICIs therapies. Can you please discuss more on these contradictory roles of these cytokines?
- in 2.3. Mechanical microenvironment, Piezo1 is a mechanosensitive cation channel expressed on both CD4+and CD8+T cells. There are papers reporting T cells sensing mechanical force through Piezo1. Can you discuss how mechanical force in TME may directly affect T cell function?
Author Response
Thank you very much for your time and effort in reviewing our manuscript, and we also appreciate your recognition of our work. We have carefully analyzed your comments and suggestions, and we believe that your insightful feedback will be highly beneficial in improving the quality of the manuscript. We have now made targeted revisions and provide our responses to your comments below:
Comment 1: In Figure 1, the FDA approved the first TIL (tumor-infiltrating lymphocyte) therapy — lifileucel (Amtagvi) — on February 16, 2024, instead of 2019.
Response 1: Thank you for pointing out the critical issue of FDA approval timing. We originally labeled 2019 in Figure 1 because the FDA did grant TIL therapy LN-145 Breakthrough Therapy Designation in June of that year. Based on your suggestion, we further reviewed the relevant materials and confirmed that FDA officially approved the first TIL therapy lifileucel (Amtagvi) in February 2024. We believe that your proposed time node is more accurate and landmark, so the date has been revised to February 2024 in the latest version of Figure 1 to ensure the accuracy and authority of the time notation.
Comment 2: In 2.1. Tumor immune microenvironment, immunosuppressive cytokines, including IL4 and IL10 were discussed. However, recently IL4 and IL10 were reported to enhance CD8 functions in TME and therefore enhanced the efficacy of CAR-T and ICIs therapies. Can you please discuss more on these contradictory roles of these cytokines?
Response 2: We sincerely appreciate your critical insight into the dual and seemingly contradictory role of cytokines such as IL-4 and IL-10 in the TME. This is indeed a delicate and fast-growing field of research. In response, we have significantly expanded the discussion in Section 2.1 (Tumor Immune Microenvironment). We clearly illustrate this duality and cite recent key studies showing that fusion protein forms of these cytokines enhance CD8+ T cell function and synergize with immunotherapy. We further propose the key insight that engineering IL-4 and IL-10, such as constructing Fc fusion proteins to prolong half-lives could potentially harness their ability to enhance anti-tumor immunity.
Comment 3: In 2.3. Mechanical microenvironment, Piezo1 is a mechanosensitive cation channel expressed on both CD4+and CD8+ T cells. There are papers reporting T cells sensing mechanical force through Piezo1. Can you discuss how mechanical force in TME may directly affect T cell function?
Response 3: Thank you for your important advice. Following your comments, we enriched the discussion in Section 2.3 (Mechanical Microenvironment). We now describe how Piezo1 acts directly on T cells by sensing elevated matrix stiffness, triggering calcium influx and downstream signaling pathways that may promote T cell depletion and upregulate inhibitory receptors such as PD-1. We also introduce recent findings suggesting that Piezo1 activation impairs T cytoskeletal dynamics and migration capabilities and explore its potential as a therapeutic target for restoring T cell cytotoxic potential.
Reviewer 2 Report
Comments and Suggestions for Authors In the review manuscript titled "Soil and Seed: Tumor Microenvironment Nurtures Immuno-therapy Resistance and Renewal” by Li et al, the authors review how the tumor microenvironment (TME) drives resistance to cancer immunotherapies such as immune checkpoint inhibitors (ICIs) and CAR-T cell therapy. They analyzed the roles of immune, metabolic, mechanical, and microbial components in mediating this resistance. Further, they discussed emerging strategies to overcome these barriers and enhance immunotherapy effectiveness. While the review is ambitious and addresses a novel aspect in Immunotherapy, there are several points to consider critically before the manuscript could be considered for publication.1) There are some spelling mistakes and typing errors that authors need to correct
2. In context of section 2.4 Microbiota Niche, there are missing fungal and viral components, which are increasingly recognized.
3) The statement that "glucose restriction is not universal" is important, however, the authors could contextualize it better and it would be helpful to clarify under which conditions this paradigm is challenged.
4) In the context of Metabolic Microenvironment, there is overemphasis on Lactate. While lactate is important, others (e.g., polyamine metabolism, tryptophan/kynurenine pathway) that are mentioned but not given equal depth. 5) In the context of microbial niche the "Dual Role" is under-explained. It correctly states that microbes have dual roles but does not sufficiently explain why so. What are the specific host factors, tumor types, or local micro-environments that determine whether Fusobacterium is pro- or anti-tumor?
6) While the CAR-T Section is heavily focused on hematological malignancies and aptly so because this is where CAR-T has had the most success. However, the discussion of resistance in solid tumors is less detailed and could benefit from more examples beyond the IFN-γR paradox.
7) The introduction is short and lacks information regarding the background of the subject. The authors need to discuss more about CAR T cell antigens involved and cite the following manuscript to enrich the literature about the CAR T cells and make the manuscript better suited for publication.
https://link.springer.com/article/10.1007/s43152-024-00055-4
8). Authors should highlight their findings in the conclusion section, as well as provide some insight on the possible therapeutic applications of their findings.
Author Response
Thank you very much for your time and effort in reviewing our manuscript, and we also appreciate your recognition of our work. We have carefully analyzed your comments and suggestions, and we believe that your insightful feedback will be highly beneficial in improving the quality of the manuscript. We have now made targeted revisions and provide our responses to your comments below:
Comment 1: There are some spelling mistakes and typing errors that authors need to correct
Response 1: We sincerely thank you for your careful review and for pointing out errors in this article. We have carefully proofread the text and corrected all identified spelling and typographical errors to ensure that the language quality of the text is of a high standard.
Comment 2: In context of section 2.4 Microbiota Niche, there are missing fungal and viral components, which are increasingly recognized.
Response 2: Thank you for your valuable suggestions to broaden the scope of the section on microbial niches. As recommended, we have supplemented Section 2.4 with a discussion of the role of fungal components and virusesin shaping TME and mediating resistance to immunotherapy, and have added descriptions of their specific mechanisms of action and interactions between microbial communities to provide a more comprehensive overview of the microbiota within tumor.
Comment 3: The statement that "glucose restriction is not universal" is important, however, the authors could contextualize it better and it would be helpful to clarify under which conditions this paradigm is challenged.
Response 3: Following your suggestion, we revised the relevant paragraphs in Section 2.2.2 to better illustrate the idea that glucose restriction is not universal. We further demonstrate that nutrient uptake levels in TME are cell-type specific and regulated by intrinsic metabolic programs such as the mTORC1 signaling pathway. In particular, it was pointed out that in tumor with high glycolytic activity in myeloid cells or where the cancer cells themselves are more dependent on alternative nutrients such as glutamine, this traditional paradigm is challenged and the pattern of glucose competition is altered.
Comment 4: In the context of Metabolic Microenvironment, there is overemphasis on Lactate. While lactate is important, others (e.g., polyamine metabolism, tryptophan/kynurenine pathway) that are mentioned but not given equal depth.
Response 4: We agree with you that a more balanced exposition of key metabolic pathways is needed. To this end, we strengthen our discussion of other important metabolic pathways in Section 2.2, including amino acid metabolism (e.g., glutamine), the tryptophan kynurenine metabolic axis, and lipid metabolism, and explore in greater depth their mechanistic roles in driving Treg differentiation, CD8+ T cell depletion, and dendritic cell dysfunction. This revision ensures that in addition to the central role of lactic acid, other key metabolic axes receive commensurate attention and depth.
Comment 5: In the context of microbial niche the "Dual Role" is under-explained. It correctly states that microbes have dual roles but does not sufficiently explain why so. What are the specific host factors, tumor types, or local micro-environments that determine whether Fusobacterium is pro- or anti-tumor?
Response 5: Thank you for this important point, which allows us to elaborate further on this section. We expanded our discussion of the "dual role" of microbiota in Section 2.4 by now analyzing in more detail factors that may contribute to its opposite effect, such as specific tumor types, local immune environments, differences in the overall composition of the microbiome and its major metabolites, thus providing a more nuanced explanation of the environmental dependence of microbial influence.
Comment 6: While the CAR-T Section is heavily focused on hematological malignancies and aptly so because this is where CAR-T has had the most success. However, the discussion of resistance in solid tumors is less detailed and could benefit from more examples beyond the IFN-γR paradox.
Response 6: In response to this comment, we enhanced the CAR-T sections (2.1.2, 2.2.2, 2.3.2) to include more examples of resistance mechanisms specific to solid tumors. These new cases include but are not limited to: tumor-related vascular system hinders CAR-T cell infiltration into solid tumor tissue, metabolic pressure, such as nutritional restriction, inhibits CAR-T function in solid tumor, etc., and further compares the similarities and differences between CAR-T resistance mechanism and blood tumor, thus providing a broader and more representative perspective for understanding CAR-T therapy in solid tumor.
Comment 7: The introduction is short and lacks information regarding the background of the subject. The authors need to discuss more about CAR T cell antigens involved and cite the following manuscript to enrich the literature about the CAR T cells and make the manuscript better suited for publication.
https://link.springer.com/article/10.1007/s43152-024-00055-4
Response 7: We appreciate your suggestions and references. We have expanded the introduction as you suggested to provide a more comprehensive background on cancer immunotherapy. Specifically, we added a discussion of the range of target antigens for CAR-T cell therapy in the areas of hematological oncology, solid tumors, and even certain non-neoplastic diseases, and cited your recommended literature to enhance the discussion in this section and recognize the important contributions in this field.
Comment 8: Authors should highlight their findings in the conclusion section, as well as provide some insight on the possible therapeutic applications of their findings.
Response 8: Thank you for your valuable advice. We have revised the conclusion section (now 4.Discussion) to provide greater clarity on the key mechanistic insights revealed in this review. In addition, we have added a new section to explore in depth the potential translational value of these mechanisms, including how to regulate TME remodeling in multiple dimensions by targeting intersections of key subsystem networks of the TME, thereby restoring sensitivity to immunotherapy. Meanwhile, we explore the feasibility and advantages of integrating multi-omics data using multidimensional convolutional neural network technology to construct personalized treatment strategies, providing new ideas for achieving breakthrough progress in the field of reversing immunotherapy resistance.
Round 2
Reviewer 2 Report
Comments and Suggestions for Authors
The authors have addressed all my concerns and have made suggested changes in the manuscript. I, therefore, recommend the manuscript suitable for publication in its revised form.